# Notch Signaling Regulates Mouse Perivascular Adipose Tissue Function via Mitochondrial Pathways

**DOI:** 10.3390/genes14101964

**Published:** 2023-10-20

**Authors:** Chenhao Yang, Xuehui Yang, Anne Harrington, Christian Potts, Abigail Kaija, Larisa Ryzhova, Lucy Liaw

**Affiliations:** 1Center for Molecular Medicine, MaineHealth Institute for Research, Scarborough, ME 04074, USA; chenhao.yang@maine.edu (C.Y.); xuehui.yang@mainehealth.org (X.Y.); anne.harrington@mainehealth.org (A.H.); christian.potts@mainehealth.org (C.P.); abigail.kaija@mainehealth.org (A.K.); larisa.ryzhova@mainehealth.org (L.R.); 2Graduate School of Biomedical Science and Engineering, University of Maine, Orono, ME 04469, USA

**Keywords:** Notch signaling, mitochondrial dynamics, perivascular adipose tissue

## Abstract

Perivascular adipose tissue (PVAT) regulates vascular function by secreting vasoactive substances. In mice, Notch signaling is activated in the PVAT during diet-induced obesity, and leads to the loss of the thermogenic phenotype and adipocyte whitening due to increased lipid accumulation. We used the Adiponectin-Cre (*Adipoq-Cre*) strain to activate a ligand-independent Notch1 intracellular domain transgene (*N1ICD*) to drive constitutive Notch signaling in the adipose tissues (*N1ICD;Adipoq-Cre*). We previously found that constitutive activation of Notch1 signaling in the PVAT phenocopied the effects of diet-induced obesity. To understand the downstream pathways activated by Notch signaling, we performed a proteomic analysis of the PVAT from control versus *N1ICD;Adipoq-Cre* mice. This comparison identified prominent changes in the protein signatures related to metabolism, adipocyte homeostasis, mitochondrial function, and ferroptosis. PVAT-derived stromal vascular fraction cells were derived from our mouse strains to study the cellular and molecular phenotypes during adipogenic induction. We found that cells with activated Notch signaling displayed decreased mitochondrial respiration despite similar levels of adipogenesis and mitochondrial number. We observed variable regulation of the proteins related to mitochondrial dynamics and ferroptosis, including PHB3, PINK1, pDRP1, and the phospholipid hydroperoxidase GPX4. Mitochondria regulate some forms of ferroptosis, which is a regulated process of cell death driven by lipid peroxidation. Accordingly, we found that Notch activation promoted lipid peroxidation and ferroptosis in PVAT-derived adipocytes. Because the PVAT phenotype is a regulator of vascular reactivity, we tested the effect of Notch activation in PVAT on vasoreactivity using wire myography. The aortae from the *N1ICD;Adipoq-Cre* mice had increased vasocontraction and decreased vasorelaxation in a PVAT-dependent and age-dependent manner. Our data provide support for the novel concept that increased Notch signaling in the adipose tissue leads to PVAT whitening, impaired mitochondrial function, increased ferroptosis, and loss of a protective vasodilatory signal. Our study advances our understanding of how Notch signaling in adipocytes affects mitochondrial dynamics, which impacts vascular physiology.

## 1. Introduction

Cardiovascular disease (CVD) remains a major cause of morbidity and mortality in the USA [1]. The rising prevalence of cardiovascular diseases (CVDs) is largely attributed to the obesity epidemic observed over the past few decades [2]. White and brown adipose tissue are two major adipose tissue types that have different molecular and cellular phenotypes. Recently, adipose tissue browning has been considered a promising strategy to treat obesity and metabolic disease. In brown adipose tissue (BAT), non-shivering thermogenesis being activated by mitochondrial uncoupling proteins (e.g., UCP1) increases heat generation and energy expenditure. BAT is thus protective against metabolic disease [3]. The perivascular adipose tissue (PVAT) surrounding large vessels in humans and thoracic PVAT in mice have a BAT-like phenotype and contain thermogenic adipocytes that produce UCP1 [4,5]. As a local vascular adipose depot, PVAT regulates vascular function via the secretion of adipokines such as adiponectin and paracrine factors to mediate vasorelaxation [6]. During obesity and vascular pathologies such as aneurysm, adipocytes in PVAT hypertrophy and increase their lipid storage, affecting their secretory function [7,8]. Thus, defining pathways important in regulating PVAT function will provide insight into vascular health.

The Notch signaling pathway regulates adipocyte homeostasis and global metabolism in mice [9]. Constitutive activation of Notch signaling in white adipose tissue was associated with reduced energy expenditure and impaired glucose homeostasis, and loss of Notch1 led to the beiging of white adipose tissue. Our group subsequently studied PVAT, and found increased Notch signaling in mouse PVAT in a diet-induced obesity model. Furthermore, constitutive activation of Notch signaling in the adipose tissue of mice fed a chow diet phenocopied the effects of high-fat diet feeding, leading to increased lipid accumulation and the whitening of PVAT [8]. Because of the impact of the PVAT phenotype on cardiovascular disease (recently reviewed [10,11]), we further studied the molecular effects of Notch activation in PVAT.

Mitochondrial pathways are closely linked to the homeostasis of adipose tissue. Impaired oxidative phosphorylation (OxPhos) pathways or mitochondrial respiration in white adipose tissue are hallmarks of obesity [12]. The pathways that control mitochondrial quality, including mitophagy, regulate beige-to-white transition in adipocytes. Mitochondria also play essential roles in regulated cell death pathways such as ferroptosis, which induces insulin resistance in adipose tissue and is tightly associated with type II diabetes mellitus [13]. Our identification of proteomic signatures in PVAT from mice with activated Notch signaling compared to controls identified differentially regulated proteins that regulate OxPhos, mitophagy, and ferroptosis. In this study, we further examined the mitochondrial pathways that are regulated by Notch signaling in PVAT to understand its impact on vascular function.

## 2. Materials and Methods

Mouse strains. All protocols using mice were approved by the Institutional Animal Care and Use Committee of MaineHealth Institute for Research. See Table A1 in the Appendix B for additional information on mouse strains. We used a Cre-LoxP strategy to generate mice with conditional Notch activation in adipose tissue. The *N1ICD* (Notch1 intracellular domain) transgenic mouse strain was generated with an FVB/N background as previously described [14]. Briefly, a floxed GFP coding sequence followed by a Myc-tagged mouse Notch1 intracellular domain (amino acids 1810-2556) was cloned downstream of an enhancer/chicken β-actin promoter to obtain a conditional *N1ICD* transgenic mouse. To activate *N1ICD* expression in the adipose tissue, homozygous *N1ICD* mice were bred with *Adiponectin-Cre* (*Adipoq-Cre*) mice (JAX, strain 028020). The F1 mice were toe-clipped at one week of age to collect genomic DNA for PCR genotyping for Cre and GFP. The PCR primers were: 

Cre forward: GCATTACCGGTCGATGCAACGAGTG 

Cre reverse: GAACGCTAGAGCCTGTTTTGCACGTT 

GFP forward: CTCGAGCCACCATGAGTAAAGGAGAAGAAC 

GFP reverse: GAACGCTAGAGCCTGTTTTGCACGTT

For the experiments, mice with the *N1ICD* transgene and Cre-negative (referred to as *N1ICD)* were used as controls, and littermates with both the *N1ICD* and Cre transgenes (referred to as *N1ICD;Adipoq-Cre)* were the experimental mice. Expression of the *N1ICD* transgene was verified via immunoblotting with an anti-Myc antibody to detect the Myc epitope tag on the *N1ICD* transgenic protein. For the high-fat diet experiments, mice were weaned at 3 weeks of age, singly housed at 7 weeks of age, and fed ad libitum with a chow diet (2018 Teklad Global 18% Protein Rodent Diet) or high-fat diet (HFD, D12492 Research Diets, New Brunswick, NJ, USA) for 12 weeks starting at 8 weeks of age. The HFD contained 60% kcal fat, 20% kcal carbohydrate, and 20% kcal protein. As female mice were resistant to weight gain on a HFD [15], only the male mice were fed with a HFD in this study. The *mito*-*QC* mice [16] were constructed with a CAG promoter and open reading frame for the mCherry-GFP-FIS1 protein (residues 101–152, an outer mitochondrial membrane binding sequence) inserted at the ROSA26 locus, and were on a C57BL6/J background. Both male and female mice were used for this study.

Proteomics and data analysis. ProteoExtract kits (Millipore Sigma, Burlington, MA, USA) were used to tryptically digest the PVAT tissue from *Adipoq-Cre* and *N1ICD* male mice on a chow diet or HFD. Peptides were separated using nanoscale liquid chromatography. SWATH was used for data acquisition and ProteinPilot was used for library construction. The methods for peptide identification and quantification were described previously [8]. Ingenuity pathway analysis (IPA, QIAGEN, Germantown, MD, USA) was used to analyze the top differentially expressed proteins with 20% change as a cut-off. Using the canonical pathway analysis, we listed the top pathways ranked by log(*p*-value). Differentially expressed proteins were subjected to STRING network and Monarch analysis of the mammalian phenotypes for proteins within networks. The mammalian phenotypes were ranked by strength, which equals log10 of the observed proteins in the corresponding network divided by the expected proteins in a random network. The lollipop plots and heatmap were generated using R.

Protein isolation and immunoblotting. Mouse PVAT was homogenized with RIPA lysis buffer (150 mM NaCl, 1% NP-40, 0.5% sodium deoxycholate, 0.1% SDS, and 50 mM Tris pH = 8.0) supplemented with 1× protease/phosphatase inhibitor cocktail (Cell Signaling) on ice. The tissue lysates were centrifuged at 1100× *g* for 20 min at 4 °C. The proteins were precipitated using 4 volumes of ice-cold 100% acetone, incubated at −20 °C for at least 1 h, pelleted using centrifugation at 10,000× *g* for 10 min at 4 °C, washed 2 times in 70% acetone, and air-dried. The pellets were resuspended in RIPA buffer supplemented with 1% SDS and sonicated at 30 watts for 10 pulses on ice using a Branson Sonifier 250. The adipocytes differentiated from the PVAT-derived stromal vascular fraction (SVF) were washed 2 times in PBS and lysed with RIPA buffer supplemented with 1× protease/phosphatase inhibitor. The lysates were left on a shaker at 4 °C for 30 min and centrifuged at 11,000× *g* for 20 min at 4 °C to collect supernatants containing protein. The protein concentrations were quantified using the Bio-Rad DC protein assay, and the protein samples were diluted with 6× Laemmli sample buffer supplemented with 100 mM dithiothreitol, and boiled at 100 °C for 3–5 min. Then, 40 to 50 μg of the protein samples were subjected to SDS-PAGE and immunoblot analysis using antibodies against UCP-1, PGC1-α, TFAM, COXIV, PHB2, PHB1, PINK1, OPA1, DRP-1, pDRP-1 (s616), GPX4, RBPJ, Myc-tag, V5-tag, and housekeeper proteins, including tubulin, cyclophilin A (CYPA), and actin (see Table A2 in Appendix B). In addition, Ponceau S staining of the total proteins was used to normalize the immunoblot results.

Primary cell culture and differentiation. The PVAT from the *N1ICD;Adipoq-Cre*, *N1ICD* (control transgenic without Cre) and *mito*-*QC* mice was digested with collagenase I (1 mg/mL) and lipase (2 mg/mL) at 37 °C for 45 min on a shaker at 300 rpm. PVAT from at least 5 mice of the same genotype and gender was pooled together for tissue processing and digestion. The digested cells were then filtered through a 70 μm nylon membrane. The filtered cells were centrifuged at 1200× *g* for 8 min, collected, and plated on 12-well plates pre-coated with 0.2% gelatin. At passage 3 or 4, the PVAT-derived SVFs were differentiated using an adipocyte induction medium containing 0.5 mM IBMX, 5 μM dexamethasone, 125 μM indomethacin, 170 nM insulin, 1 nM T3, 1 μM rosiglitazone, 5 μM TGF-β RI Kinase Inhibitor VI, and 50 μg/mL AA2P in DMEM/F12 50/50 supplemented with 10% FBS for 3 days and then changed to a maintenance medium containing 170 nM insulin, 1nM T3, 1 μM rosiglitazone, 5 μM TGF-β RI Kinase Inhibitor VI, and 50 μg/mL AA2P in DMEMF12 50/50 supplemented with 10% FBS for 4 days.

Seahorse assay. The oxygen consumption rate (OCR) and extracellular acidification rate (ECAR) were assessed using the XFe96 Extracellular Flux Analyzer (Seahorse Biosciences, North Billerica, MA, USA). PVAT-derived stromal vascular fraction cells were isolated from the *N1ICD;Adipoq-Cre* or *N1ICD* control mice, and were plated at a density of 5000 cell/well. After 24 h, the cells were differentiated for 6–7 days in 96-well Seahorse XF96 cell culture microplates before chemical treatment including mitochondrial complex inhibitors (2 μM oligomycin, 2 μM rotenone, and 2 μM antimycin) and 2 μM of the uncoupler of oxidative phosphorylation, FCCCP. The cell numbers were used for the normalization of the data and counted after Hoechst staining.

RT-qPCR and mitochondrial copy number assay. The total RNA from the differentiated PVAT-derived SVFs was extracted using TRIzol Reagent (Thermo Fisher, Waltham, MA, USA) and the total RNA was isolated using the RNeasy Plus Mini Kit (QIAGEN, Germantown, MD, USA). The RNA quality and quantity were assessed using the Nanodrop (Thermo Fisher, Waltham, MA, USA) and Bioanalyzer RNA Nano assay (Agilent Technology, Santa Clara, CA, USA). RNA samples with a RIN# 7 and above were used in this study. The cDNA was synthesized using the AzuraQuantTM cDNA synthesis kit, and the qPCR reactions were performed using the AzuraQuant Green Fast qPCR master mix. The relative mRNA levels were calculated using the comparative CT method with *Ppia* or *Actb* as reference genes. The primer sequences are listed in Table A3 in Appendix B. For mitochondrial DNA isolation, the PVAT was homogenized in lysis buffer supplemented with 0.2 mg/mL proteinase K at 55 °C overnight and further isolated using 7.5 M ammonium acetate and 70% isopropanol as described previously [17]. The mitochondrial DNA was suspended in TE buffer and the relative copy number of the mitochondrial DNA was quantified using the comparative CT method with nuclear-encoded 18s rRNA as the normalizer.

Cell staining and confocal microscopy imaging. For the detection of lipid peroxidation, the PVAT-derived SVF cells were differentiated on a 96-well plate with an optical bottom. Cells were washed 2 times with PBS containing 0.2% fatty acid-free bovine albumin and stained with BODIPY™ 581/591 C11 (all from Thermo Fisher, Waltham, MA, USA) for 30 min at 37 °C. The cells were washed with fatty acid-free BSA before staining with DAPI. Images were captured using a Leica SP8 confocal microscope with a 10× objective and the green and red fluorescence of BODIPY C11 was acquired using excitation wavelengths of 488 nm and 514 nm. Quantification was performed using ImageJ (NIH, version 1.53t). For Oil Red O (ORO) staining, cells differentiated from the SVF were gently rinsed with PBS and fixed with 10% formalin for 30 min at room temperature. To completely cover the cells, 60% isopropanol was added. After pouring off the isopropanol, the cells were stained with a filtered ORO working solution at a final 0.3% ORO (obtained by mixing 0.5% Oil Red O in 60% isopropanol with ddH_2_O at a ratio of 3:2) for 5 min at room temperature. The stained cells were rinsed with tap water gently and imaged under brightfield using a Canon EOS 60D camera and a Zeiss microscope (Axiovert 40c) using a 10× objective lens with phase contrast. For the quantification of ORO staining, the stained cells were dried overnight, eluted with 100% isopropanol, and quantified using a BioTek Epoch plate reader at 490 nm. After that, the cells were washed with water and stained with 0.05% crystal violet for 10 min at room temperature. The crystal violet dye was eluted with 10% acetic acid and quantified at 600 nm. The ORO stain was normalized to the crystal violet stain.

Vessel wire myography. The wire myography experiments were conducted using a DMT Multiple Myography System (620 M, DMT USA, Ann Arbor, MI, USA). On the day of the experiment, the mice were intraperitoneally injected with 300 units of heparin and euthanized. A 2 mm anatomically standardized segment of the thoracic aorta from each mouse was mounted in a myography chamber filled with PSS buffer (130 mM NaCl, 4.2 mM KCl, 1.18 mM KH_2_PO_4_, 1.17 mM MgSO_4_.7H_2_O_,_ 24.9 mM NaHCO_3_, 5.5 mM glucose, 0.026 mM EDTA, and 1.6 mM CaCl_2_) at 37 °C with constant aeration with 5% CO_2_ and 95% O_2_. The aortae were equilibrated in the chamber for 20 min until stabilization. The zeroing and normalization steps were performed according to the manufacturer’s instructions. The vessels were stimulated with 100 mM KCl for 8 min until contraction plateaued. After washing, the vessels were relaxed and stabilized. Increasing doses of phenylephrine (2 nM–100 μM, Millipore Sigma, Burlington, MA, USA) were then applied to induce vascular contraction. To assess vascular relaxation, the aortae were precontracted with phenylephrine to ~50% of maximum contractibility as determined in the first round of phenylephrine treatment. The precontracted vessels were treated with increasing concentrations of acetylcholine (2 nM–100 μM, Millipore Sigma, Burlington, MA, USA). Finally, the vessels were treated with 100 mM KCl for 8 min to assess their viability at the end of the experiment, and the data were recorded. Data from vessels that did not respond to the final KCl treatment were not included in subsequent analyses. Vascular contraction to the phenylephrine dosages was represented as the percentage of KCl-induced contraction (% KCl). Vascular relaxation to acetylcholine was calculated as the percentage of phenylephrine-induced pre-contraction (% Phe). To compare the contraction and relaxation responses of each group, the dose response curves were fitted using a non-linear regression model with the options of log(agonist/inhibitor) vs. response-variable slope (four parameters) in Prism. The logEC50 of each group was statistically compared using an extra sum-of-squares F test and the statistical significance was calculated.

Statistical analysis. GraphPad Prism 9 was used for the statistical analysis. An unpaired Student’s *t*-test was used to compare two groups of data. Two-way ANOVA was used to compared more than two groups of data. Results were considered significantly different with *p* < 0.05, and all data were reported as means ± SEM.

## 3. Results

### 3.1. Notch Activation Regulates Metabolism-Related Pathways in Perivascular Adipose Tissue In Vivo

We generated mice with constitutively activated Notch signaling in their adipocytes [8] by crossing conditional homozygous *N1ICD* transgenic mice with heterozygous *Adipoq-Cre* mice (Appendix A). We obtained a total of ~301 animals (150 females and 151 males), and the numbers of Cre-negative *N1ICD* and double-positive *N1ICD;Adipoq-Cre* mice were 159 and 142, roughly ~50% for each genotype. The *N1ICD;Adipoq-Cre* mice appeared normal and healthy compared to their *N1ICD* littermates controls on a regular chow diet. We tested whether the *N1ICD;Adipoq-Cre* mice had changes in their perivascular adipose tissue (PVAT) when the mice were fed either a chow diet or a high-fat diet (HFD, 60 kcal% fat, Appendix A). Via hematoxylin and eosin staining (H&E) of the PVAT sections, we observed increased lipid accumulation in the PVAT from the *N1ICD;Adipoq-Cre* male mice fed a chow diet, compared to the *N1ICD* littermate controls (Appendix A), consistent with our previous report [8], and HFD feeding further increased this effect. The lipid content in the PVAT from female mice was lower compared to the PVAT from male mice. In contrast to the PVAT in male mice, there were no significant differences in lipid accumulation in the PVAT between female *N1ICD;Adipoq-Cre* mice and their littermate *N1ICD* controls (Appendix A).

PVAT is a metabolically active tissue, and in mice, has a similar morphology to brown adipose tissue. To define the potential pathways involved in the lipid changes in PVAT induced by Notch signaling, we performed quantitative proteomics on the PVAT from *N1ICD;Adipoq-Cre* and *N1ICD* control mice on a regular chow diet or after 12 weeks of HFD feeding. It is well known that female mice are resistant to high-fat-diet-induced obesity. Because we also observed that the PVAT from female *N1ICD;Adipoq-Cre* mice had no histopathological differences compared to *N1ICD* littermate controls, we focused this analysis on male mice. Differences in proteomic signatures were identified based on genotype on both a standard chow diet and on a HFD. In the mice maintained on a standard chow diet, the PVAT from *N1ICD;Adipoq-Cre* mice showed a decrease in 100 proteins, including mitochondrial complex proteins, and an increase in 318 proteins, including proteins involved in fatty acid oxidation (Figure 1A).

Several notable changes included an increase in FHL1, which is involved in the stress response to aortic constriction [18]. We also found that mitochondrial proteins including PHB2, IPYR2, SDHA, and TIM10 were differentially expressed in PVAT with activated Notch signaling. In addition, the antioxidant protein PRDX2 was significantly lower in the PVAT with the activation of Notch signaling. PRDX2 is involved in the inhibition of ferroptosis, an iron-dependent cell death process driven by lipid peroxidation that is associated with mitochondrial dysfunction [19]. We performed Ingenuity Pathway Analysis of the significantly different proteins between our experimental groups, and found that mitochondrial dysfunction, oxidative phosphorylation, glycolysis, and fatty acid oxidation were among the significantly regulated pathways (Figure 1B). The mammalian phenotype ontology analysis of the differentially expressed proteins within STRING protein networks indicated that Notch activation affects PVAT homeostasis/metabolism and mitochondrial function (Figure 1C and Figure 2B).

In our diet-induced obesity model, activation of Notch signaling in PVAT led to a decrease in 228 proteins and an increase in 142 proteins (Figure 1D). In mice fed for 12 weeks with a HFD, activated Notch signaling led to a decrease in the mitochondrial complex proteins that regulate respiration, including COX7C (Figure 1D). Ingenuity Pathway Analysis showed that the mitochondrial-dysfunction- and oxidative-phosphorylation-associated pathways were also significantly altered by Notch activation in mice on a HFD (Figure 1E), suggesting that constitutive activation of Notch signaling impairs mitochondrial respiratory function. In addition, fatty acid metabolism and triglyceride levels were identified as phenotype terms associated with significantly different proteins in the PVAT from *N1ICD* versus *N1ICD;Adipoq-Cre* mice fed a HFD for 12 weeks (Figure 1F). These results are consistent with changes in metabolism in PVAT as drivers of the lipid phenotype observed.

We then independently compared the effects of diet or genotype using an analysis of the mammalian phenotype terms. Proteins that were different in the control *N1ICD* mice fed a HFD versus control chow clustered in terms related to adipose, metabolism, and mitochondrial phenotypes (Figure 2A). In an analysis of *N1ICD* versus *N1ICD;Adipoq-Cre* mice fed a chow diet, adipose, metabolism, mitochondrial, and cardiac phenotype terms were represented (Figure 2B). Because of the consistent finding of metabolism and mitochondrial terms, we identified the proteins contributing to these categories and generated heatmaps to visualize their protein levels comparing *N1ICD;Adipoq-Cre* versus *N1ICD* either on a chow diet or HFD (Figure 2C). The patterns show that both diet and genotype significantly affect a significant complex of proteins related to metabolism and mitochondrial phenotypes.

### 3.2. Activation of Notch Signaling Alters Mitochondria Biogenesis and Metabolism Gene Expression in Differentiated PVAT SVFs

The alteration of mitochondrial and metabolism protein signatures in vivo due to the activation of Notch signaling prompted us to study mitochondrial respiration. We isolated the PVAT to obtain the stromal vascular fraction (PVAT-SVF) from *N1ICD;Adipoq-Cre* and *N1ICD* mice at 8 weeks of age and subjected the primary cultures to adipogenic differentiation (see Table A4, Table A5 and Table A6 in Appendix B for cell culture media). A Seahorse Cell Mito Stress Test was then performed on the PVAT-SVF differentiated adipocytes. Because the activation of Notch in the PVAT of male versus female mice in vivo had different effects on the PVAT morphology and lipid content (Appendix A), we analyzed the PVAT-SVFs from male or female mice separately. Unexpectedly, Notch activation resulted in similarly reduced mitochondrial respiration in the differentiated adipocytes derived from the PVAT-SVFs from both male and female mice (Figure 3A,B). Maximal respiration was suppressed in the PVAT-derived adipocytes from both sexes from the *N1ICD-Adipoq-Cre* mice, and female mice additionally had lower ATP production.

It is known that adipogenic differentiation also influences mitochondrial respiration, and Notch1 signaling regulates the process of browning in white adipose tissue. To determine whether changes in adipogenic capacity were responsible for the observed changes in cellular bioenergetics, we examined the adipogenic capacity in the PVAT-SVF from the *N1ICD;Adipoq-Cre* and *N1ICD* mice. There was a similar level of adipogenic differentiation (Appendix A). The lipid accumulation measured using Oil Red O staining normalized to cell number was similar regardless of genotype in the PVAT-SVF differentiated adipocytes from male or female mice (Appendix A). Parallel experiments utilized immunoblot and RT-qPCR analysis to detect perilipin1 (PLIN1), a lipid droplet membrane protein. There were no significant differences in the Plin1 protein levels between the male and female *N1ICD;Adipoq-Cre* and *N1ICD* control PVAT-derived differentiated adipocytes (Appendix A). These results suggest that the activation of Notch in adipocytes has a selective effect on mitochondrial biogenesis in adipocytes rather than a more general effect on adipogenesis.

Next, we examined the expression of mitochondria biogenesis and metabolism-associated genes using RT-qPCR in adipocytes from the PVAT-SVF to explore the potential mechanisms downstream of Notch activation (Figure 3C). Activation of Notch signaling consistently increased adiponectin (*Adipoq*) transcripts, while mRNA levels of the mitochondrial biogenesis marker *Pgc1α* was reduced in differentiated PVAT-SVF cells from male but not female mice. Conversely, *Tfam*, *Ucp1,* and *Cidea* were significantly reduced only in the cells derived from female *N1ICD-Adipoq-Cre* mice. To determine whether these differences in vitro were reflected in the PVAT in vivo, we performed immunoblotting of whole PVAT. Analysis of UCP1, PGC1a, TFAM, and COXIV in whole PVAT using immunoblotting did not show major differences in protein levels based on genotype (Figure 3D). There are several possibilities to explain the difference between the primary adipocyte cultures and the whole PVAT tissue. Cells in vitro potentially exhibit different responses to signaling stimuli, and additionally, the PVAT whole tissue has multiple cell types contributing to the immunoblot results, unlike the primary adipocyte cultures.

To examine whether the activation of Notch signaling affected the mitochondrial number in PVAT-SVF cells, we performed a quantitative PCR for the mitochondrial DNA. The results showed no difference in the mitochondrial DNA copy numbers between either male or female *N1ICD;Adipoq*-Cre and *N1ICD* controls fed a chow diet (Figure 1F). Collectively, these data show that the overall features of the mitochondria in PVAT-SVF derived adipocytes are not overtly different, but functional bioenergetics is impaired with activated Notch signaling, suggesting a targeted regulation of mitochondrial metabolism.

### 3.3. Notch Signaling Regulates Mitochondrial Dynamics in PVAT-SVF

Mitochondrial dynamics and quality control are key for maintaining cellular health and efficient function. As mitophagy is a crucial mitochondrial quality control process, we examined the expression of mitophagy markers in vivo and in vitro. The inner mitochondrial membrane protein prohibitin 2 (PHB2) is a mitophagy receptor that regulates the PINK1-Parkin mitophagy pathway [20]. We found that protein levels of PHB2 and PINK1 were significantly upregulated in the PVAT of male *N1ICD;Adipoq-Cre* mice compared to the *N1ICD* controls maintained on a chow diet (Figure 4A,B). We also assessed the steady-state transcript levels of mitophagy regulators including *Phb2*, *Pink1*, and *Bnip3* using RT-qPCR (Figure 4C). Most of these transcripts were not significantly different between genotypes, except for *Pink1* mRNA, which was elevated in female PVAT with activated Notch signaling. We also examined proteins related to mitochondrial fusion and fission processes. We found that protein levels of the long isoforms of mitochondrial fusion regulator OPA1 were significantly decreased in the female *N1ICD;Adipoq*-Cre PVAT-SVF differentiated adipocytes compared to the controls. The phosphorylation DRP-1 (pDRP-1 at Ser166), a mitochondrial fission regulator, was increased in the female *N1ICD;Adipoq*-Cre PVAT-SVF differentiated adipocytes compared to the controls (Figure 4D,E).

To visualize the effect of Notch activation on mitophagy in adipocytes, we utilized the *mito-QC* mice that had mCherry-GFP anchored on the mitochondrial outer membrane. The PVAT-SVFs isolated from *mito-QC* mice were transduced with adenoviral constructs expressing *N1ICD* or its downstream transcriptional co-activator, RBP-Jk (Figure 4F). The cells were then treated with CCCP, an inhibitor of mitochondrial oxidative phosphorylation, to induce mitochondrial dysfunction as a positive control. CCCP treatment induced mitophagy and mitochondria degradation as indicated by reduced GFP and mCherry signals compared to the DMSO controls. Similarly, N1ICD transfection decreased the GFP and mCherry signals (Figure 4G). We also quantified the mitochondrial turnover by assessing the presence of mitolysosomes, in which the low pH environment quenches GFP, resulting in mCherry-only puncta. The results show that activated Notch signaling in the PVAT-SVF significantly increased the number of mitolysosomes per cell and per area (Figure 4G,H), whereas expression of RBP-Jk alone did not affect the mitolysosomes. These data support the concept that Notch signaling regulates mitochondrial function through, at least partly, promoting mitophagy in the PVAT-SVF.

### 3.4. Notch Signaling Regulates Ferroptosis Pathways in PVAT-SVF Differentiated Adipocytes In Vitro

Changes in mitochondrial fusion and fission can lead to mitochondrial dysfunction, increasing oxidative stress and ferroptosis [21,22]. Ferroptosis is a process of regulated necrosis that is affected by mitochondrial lipid peroxidation; proteins that protect against lipid peroxidation can inhibit ferroptosis [23]. In other organ systems, Notch has been associated with ferroptosis [24,25]. Our proteomics analysis also identified the ferroptosis pathway as a phenotype of interest, so we examined glutathione peroxidase 4 (GPX4), an antioxidant enzyme that is a negative regulator of ferroptosis pathways. We quantified the protein levels of GPX4 in the differentiated PVAT-SVF from the *N1ICD;Adipoq-Cre* mice and *N1ICD* control littermates. Activation of Notch led to a reduction in GPX4 protein in the PVAT-SVF differentiated adipocytes, which was trending in cells from male mice and highly significant in cells from female mice (Figure 5A,B). Next, we measured lipid peroxidation, which is one of the important biochemical hallmarks of ferroptosis. Differentiated adipocytes from *N1ICD;Adipoq*-Cre mice and *N1ICD* control PVAT-SVFs were stained with the lipid peroxidation sensor, BODIPY™ 581/591 C11, in which oxidation results in a shift in its excitation/emission from 581/591 nm (red fluorescence) to 481/510 nm (green fluorescence). We found increased shifting of the emission maxima of red fluorescence toward green fluorescence in the *N1ICD;Adipoq-Cre* PVAT-SVF differentiated adipocytes compared to the controls (Figure 5C,D), indicating that Notch activation promotes lipid peroxidation and ferroptosis.

### 3.5. Activation of Notch in PVAT Leads to Altered Vascular Function

Recent attention has focused on the physiological impact of the PVAT on vascular physiology [11,26,27,28,29]. The paradigm of paracrine signaling from the PVAT to the underlying vessel wall is particularly important in the regulation of vasoreactivity and vascular disease. Because of our observations that Notch activation in the adipose tissue significantly affects PVAT morphology, cellular bioenergetics, and mitochondrial function, we tested whether there was an impact on the vasoreactivity of the underlying aorta. We performed wire myography experiments using aortic segments with or without PVAT from male *N1ICD;Adipoq-Cre* and *N1ICD* mice at 8 or 20 weeks of age that had been maintained on a standard chow diet. At 8 weeks of age, the aortae with PVAT from *N1ICD;Adipoq-Cre* mice showed an increased contractile response to increasing doses of phenylephrine (2 nM–100 μM) compared to the *N1ICD* controls (Figure 6A). Correspondingly, these aortae with PVAT from mice with Notch activation in their adipocytes showed a reduced relaxation response to acetylcholine treatment (2 nM–100 μM) compared to the *N1ICD* controls (Figure 6B). Similarly, the aortae with PVAT from *N1ICD;Adipoq-Cre* mice at 20 weeks of age displayed increased contractile responses to phenylephrine (Figure 6C), although at this age, the relaxation response unexpectedly exceeded the control response.

We confirmed that the altered vasoreactivity of the aorta from the *N1ICD;Adipoq-Cre* was dependent on the PVAT. The same experiments were conducted, but the PVAT was removed from the aortic segment prior to myography. Without PVAT, the differences seen in the vessels from *N1ICD;Adipoq-Cre* were abolished (Figure 6E). Both effects on vasocontraction and vasorelaxation were PVAT-dependent, as PVAT removal resulted in the loss of these effects (Figure 6E,F). To further study the pathways involved in vasoreactivity, we analyzed the eNOS/NO pathway. Immunoblotting analysis showed that the PVAT from the 20 week old *N1ICD;Adipoq-Cre* mice had increased phosphorylation of eNOS at Ser1177 (Figure 6G,H), an active form of eNOS that produces NO. These data suggest that Notch activation in adipocytes can impact the underlying vessel by modulating vasoreactivity through eNOS activation.

## 4. Discussion

PVAT releases adipokines and bioactive molecules, including ROS and NO, to regulate vasoreactivity through paracrine interaction with the blood vessel wall. The phenotype of PVAT and its function is significantly altered during obesity, which leads to increased inflammation and vasoconstriction [11]. As an important developmental pathway, Notch signaling also regulates metabolic homeostasis and adipogenesis. Loss of Notch signaling in the adipose tissue correlates to improved thermogenesis, while overactivation of Notch in adipose tissue leads to decreased energy expenditure and impaired global metabolism [9]. Our lab previously showed that Notch signaling was significantly upregulated in the PVAT of mice fed a high-fat diet, and Notch activation in the adipose tissue leads to increased lipid accumulation and increased inflammation in the PVAT [8]. These data collectively support the hypothesis that Notch regulates PVAT physiology and its vasoregulatory function. In the current study, we report that constitutive activation of Notch signaling in the adipocytes in *N1ICD;Adipoq-Cre* mice was associated with distinct protein signatures in the PVAT that correspond to pathways involved in metabolism, mitochondrial function, and ferroptosis. Novel molecular analyses were performed to study the mechanisms and pathways activated by Notch signaling in PVAT. Major discoveries include decreased mitochondrial respiration and increased ferroptosis in PVAT-SVF differentiated adipocytes in vitro, and PVAT-dependent loss of vasodilatory effect on the aortae ex vivo, likely via regulation of eNOS phosphorylation and activity. Collectively, we found that constitutive activation of N1ICD in the adipocytes significantly altered the PVAT’s metabolism and function, and further impacted aortic function.

Our study is novel in finding a link between Notch signaling in PVAT and mitochondrial dynamics and ferroptosis. In adipose tissue, there is an interesting connection between mitochondrial respiration and adipogenesis. In human mesenchymal stem cells, adipogenic differentiation increased mitochondrial oxygen consumption with an apparent switch to oxidative phosphorylation [30]. In this case, increased mitochondrial biogenesis was associated with differentiation, and inhibition of mitochondrial oxidation suppressed adipogenesis. Thus, Notch signaling affecting mitochondrial dynamics may affect the overall health of the PVAT by changing its capacity for differentiation, affecting oxidative stress responses to generate reactive oxygen species [31], or affecting the inflammatory environment [32]. Importantly, these features of PVAT are associated with metabolic diseases including obesity and diabetes, which are major risk factors for cardiovascular disease.

Mitochondria also have a central role in ferroptosis, which involves iron-dependent lipid peroxidation, leading to a regulated form of necrosis. A previous study using a human fibrosarcoma tumor cell line indicated that mitochondria were involved in some forms of ferroptosis, specifically due to cysteine deprivation [33]. Ferroptosis is also induced by the inhibition of GPX4, which we observed was modulated by Notch signaling in PVAT-derived adipocytes. Although ferroptosis in adipose tissue has not been as highly studied as in cancer, there are several intriguing connections.

Ferroptosis is closely linked to mitochondrial dysfunction. Increased mitochondrial ROS released from an overactivated electron transport chain (ETC) could lead to increased oxidation of polyunsaturated fatty acids (PUFAs) [33] and indirectly upregulate PUFA synthesis through the inhibition of the AMPK pathway [34], leading to increased lipid peroxidation. In addition, impaired mitochondria could also downregulate GPX4 through the inhibition of the integrated stress response pathway mediated by EIF2α phosphorylation. Our in vitro experiments showed that GPX4 levels were significantly reduced by Notch activation. There was a corresponding increase in lipid peroxidation in PVAT cells with Notch activation as detected using BODIPY C11 staining. This is a novel association between Notch and the ferroptotic pathway in adipocytes. Previous studies in cancer cells found that Notch3 regulates lipid peroxidation and reactive oxygen species, and a loss of Notch3 function induced ferroptosis in non-small cell lung cancer cells [35]. It is likely that there is tissue specificity in the Notch family regulation of ferroptosis, which requires further study.

Ferroptotic cell death has been shown to play an important role in the pathogenesis of cardiovascular disease, including atherosclerosis and type II diabetes [36,37]. Therapeutic strategies targeting lipid peroxidation using a free radical scavenger such as ferrostatin-1 alleviated atherosclerosis induced in ApoE-null mice fed a HFD [38]. It will be interesting to study whether increased ferroptosis in the PVAT could lead to vascular pathology. We tested the vascular function in our mouse strains ex vivo using wire myography, and found that the aortae with PVAT dissected from *N1ICD;Adipoq*-Cre mice at 8 and 20 weeks of age showed increased vascular contraction. In general, the overactivation of Notch signaling in the PVAT could lead to increased ferroptosis-associated lipid peroxidation in the adipocytes and increased secretion of free radicals into the smooth muscle cells or endothelial layer of the PVAT-adjacent vessel. As a result, the anticontractile effects of the PVAT may be compromised while the inflammatory response is increased, promoting cardiovascular pathology.

It is worth noting the limitations of our study. We have mainly focused on the in vivo effects and functions of Notch signaling in male mice due to female mice being resistant to diet-induced obesity. However, the in vitro analyses showed that Notch signaling had effects on PVAT-SVF differentiated adipocytes derived from both male and female mice, and there were differences in response. This suggests that sex-related factors can be influential even in isolated primary cells. We have explored the mitochondrial mechanisms of Notch signaling effects on PVAT adipocytes at the levels of critical molecules. Further visualization and monitoring of the mitochondrial ultrastructure and dynamic changes regulated by Notch signaling are necessary to verify our findings.

## 5. Conclusions

In the present study, we investigate the mechanisms by which Notch signaling in PVAT affects its morphology, lipid accumulation, and regulation of vascular function. Our *N1ICD-Adipoq-Cre* mouse model effectively activates Notch signaling in the PVAT, which shows a sexually dimorphic phenotype. In vivo, male mice with activated Notch signaling have expanded lipid accumulation in the PVAT, while female mice do not. However, PVAT-derived SVF cells and differentiated adipocytes from both sexes show impaired cellular bioenergetics, increased numbers of mitolysosomes, and ferroptosis, indicating that PVAT regulates mitochondrial dynamics. Dysfunctional mitochondria in the PVAT-derived adipocytes in *N1ICD-Adipoq-Cre* mice were associated with increased ferroptosis. These changes may promote vascular pathology through the paracrine release of free radicals from the PVAT. Future studies will focus on the therapeutic potential of targeting Notch signaling in PVAT as a mechanism to decrease risk of cardiovascular disease.

## Figures and Tables

**Figure 1 genes-14-01964-f001:**
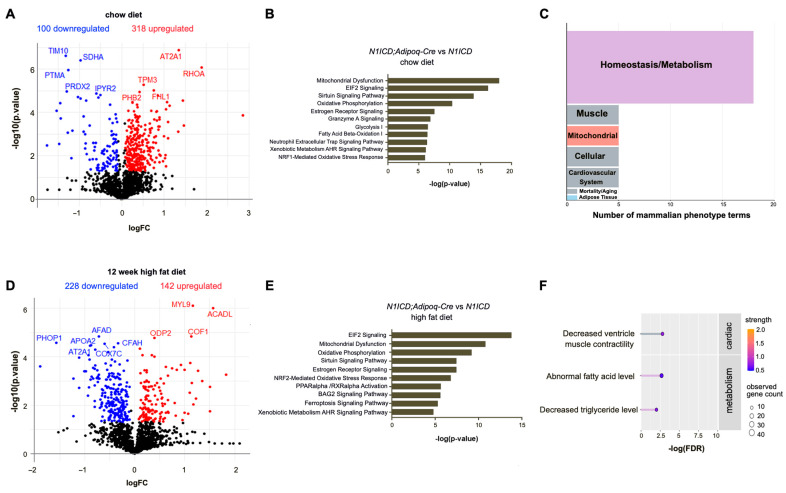
Notch activation regulates protein signatures of mitochondrial-related metabolic pathways in perivascular adipose tissue. (**A**,**D**) Volcano plots show downregulated and upregulated proteins in PVAT of *N1ICD;Adipoq-Cre* compared to control *N1ICD* mice fed with chow diet (**A**) or high-fat diet (**D**) for 12 weeks, starting at 8 weeks of age. The top 10 differentially produced proteins in PVAT ranked by significance are highlighted in each plot. (**B**,**E**) Ingenuity pathway analysis shows top differentially regulated pathways in the PVAT of *N1ICD;Adipoq-Cre* compared to PVAT of control *N1ICD* mice fed with chow diet (**B**) or high-fat diet (**E**), with 20% as the cut-off value for fold change. (**C**) Major mammalian phenotype terms of differentially expressed proteins in PVAT of *N1ICD;Adipoq-Cre* mice compared to the PVAT of *N1ICD* control mice fed with chow diet. The mammalian phenotype terms were filtered by strength (>0.5) and false discovery rate (<0.01). (**F**) The lollipop plot shows mammalian phenotype terms of differentially expressed proteins in PVAT of *N1ICD;Adipoq-Cre* mice compared to the PVAT of *N1ICD* mice fed with high-fat diet. Strength equals log10 (observed proteins in the corresponding network/expected proteins in a random network). False discovery rate (FDR) represents significance of the enrichment.

**Figure 2 genes-14-01964-f002:**
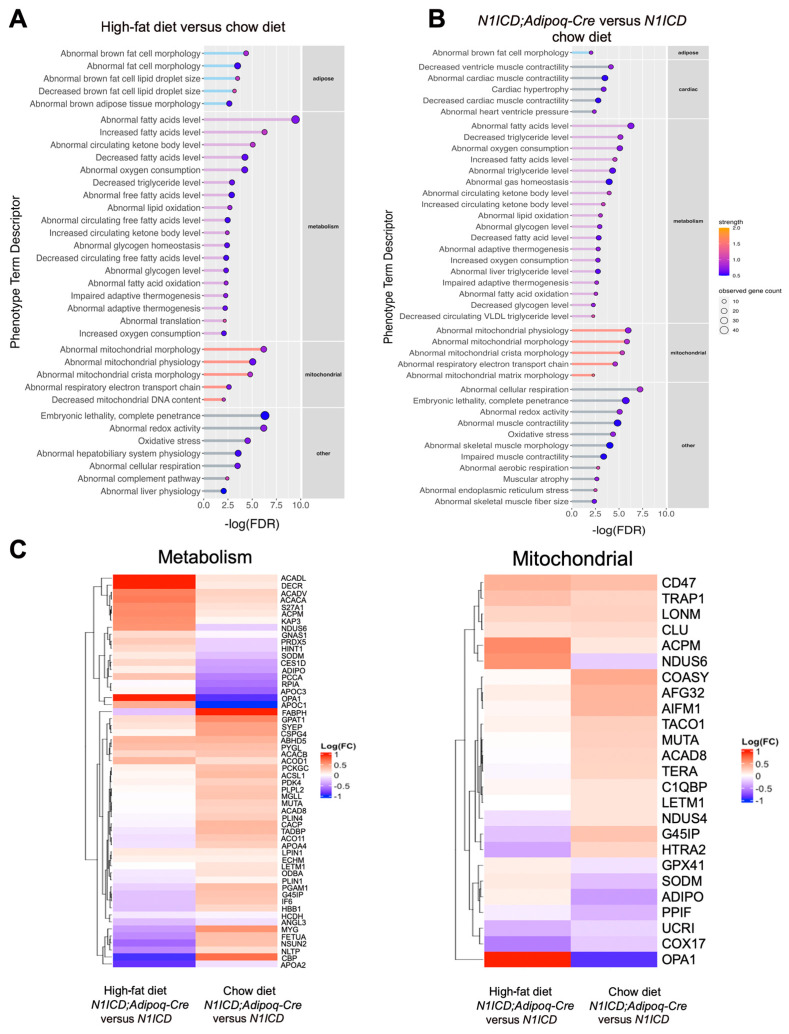
Mammalian phenotype terms of differentially expressed proteins in PVAT in response to high-fat diet and Notch activation. (**A**) Lollipop plot shows the mammalian phenotype terms of differentially expressed proteins in the PVAT of *N1ICD* mice fed with a high-fat diet compared to the PVAT of *N1ICD* mice fed with chow diet. (**B**) Lollipop plot shows the mammalian phenotype terms of differentially expressed proteins in the PVAT of *AdpoqCre;N1ICD* mice compared to the PVAT of *N1ICD* (control, no Cre) mice fed with chow diet. Strength equals log10 (observed proteins in the corresponding network/expected proteins in a random network). False discovery rate (FDR) represents significance of the enrichment. The mammalian phenotype terms were filtered by strength (≥0.5) and false discovery rate (≤0.01). (**C**) Heatmap shows proteins contributing to the mammalian phenotype term of either “Homeostasis/metabolism” or “Abnormal Mitochondrial Morphology” and “Abnormal mitochondrial physiology” (“Mitochondrial”) that were differentially expressed in the PVAT of *Adipoq-Cre;N1ICD* compared to the PVAT of *N1ICD* (control) mice fed with chow diet or high-fat diet.

**Figure 3 genes-14-01964-f003:**
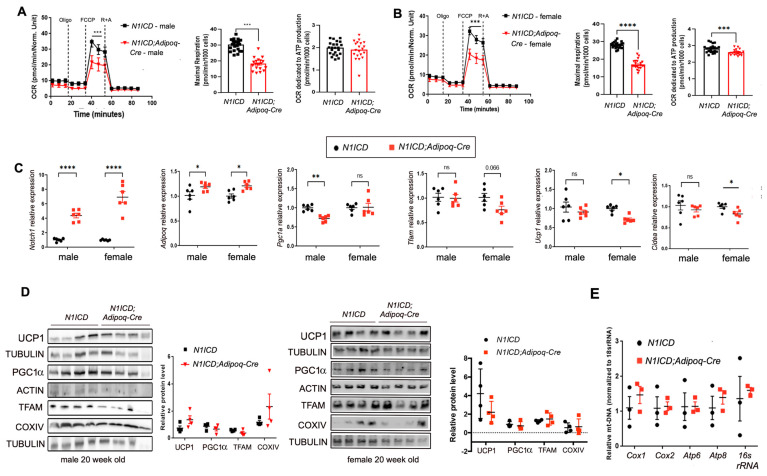
Notch signaling regulates adipocyte bioenergetics and mitochondrial biogenesis. (**A**,**B**) The oxygen consumption rate (OCR), maximal respiration, and ATP production were studied in differentiated adipocytes derived from the PVAT-SVFs of 8 week old mice. The OCR was reduced in differentiated adipocytes pooled from *N1ICD;Adipoq-Cre* male mice compared to *N1ICD* male mice (n = 6/group). (**B**) The same analysis was performed in cells derived from female mice. OCR values were normalized to cell number. (**C**) Total RNA was collected from the same experimental groups in males and females for RT-qPCR to detect markers indicated. (**D**) Immunoblotting was performed with total PVAT protein lysates from 20 week old mice of the genotypes and sex indicated. (**E**) Mitochondrial copy number for the genes indicated were assayed using qPCR and normalized to 16s rRNA by the mitochondrial DNA/nuclear DNA ratio. Graphed are means ± SEM. *p* values were determined using two-way ANOVA analysis. Statistical significance is displayed as *: *p* < 0.05; **: *p* < 0.01; ***: *p* < 0.001, ****: *p* < 0.0001.

**Figure 4 genes-14-01964-f004:**
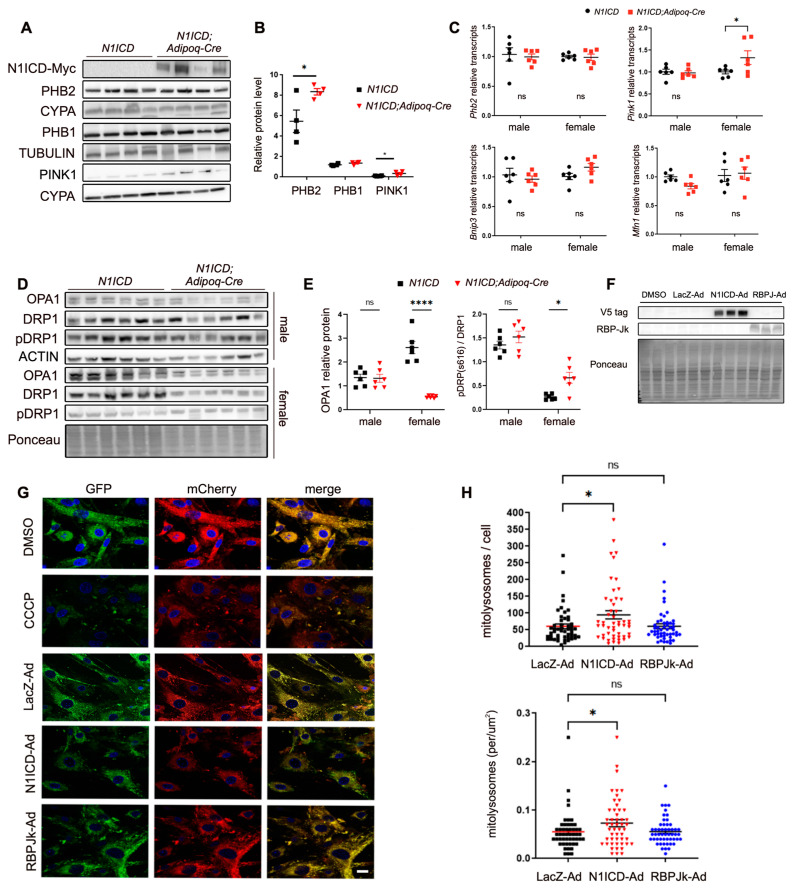
Notch signaling activation regulates mitochondrial quality control independent of mitophagy in PVAT-derived adipocytes. (**A**) Immunoblot analysis of PVAT from *N1ICD* and *N1ICD;Adipoq-Cre* 20 week old male mice maintained on a chow diet (n = 4 for each genotype). Markers of mitophagy and mitochondrial dynamics were assessed and quantified (**B**). (**C**) RT-qPCR examination of *Phb2*, *Pink1*, *Bnip3*, and *Mfn1* mRNA levels in PVAT from *N1ICD;Adipoq-Cre* and *N1ICD* control animals. (**D**) Immunoblot analysis was performed from lysates from differentiated PVAT-SVF from male and female mice of the indicated genotypes, and quantified in (**E**). (**F**,**G**) PVAT-SVFs were isolated from 8 week old *mito-QC* mice fed a chow diet for examination of the effect of activation of Notch signaling on mitophagy. Cells were transduced with N1ICD-V5 adenovirus (N1ICD-Ad), RBPJ adenovirus (RBPJ-Ad), or negative control expressing LacZ (LacZ-Ad), and treated with DMSO (negative control) or 20 μM CCCP (mitophagy inducer) for 24 h. mCherry-only foci indicate mitolysosomes. Scale bar = 20 μm. (**H**) Mitophagy levels were quantified using the mito counter plugin within ImageJ (version 1.53t), and were represented as mitolysosome number per cell and mitolysosome number/μm^2^ cell area (n = 50 cells were quantified for each group). Statistical analysis was performed using one-way ANOVA. Statistical significance is displayed as *: *p* < 0.05, ****: *p* < 0.0001.

**Figure 5 genes-14-01964-f005:**
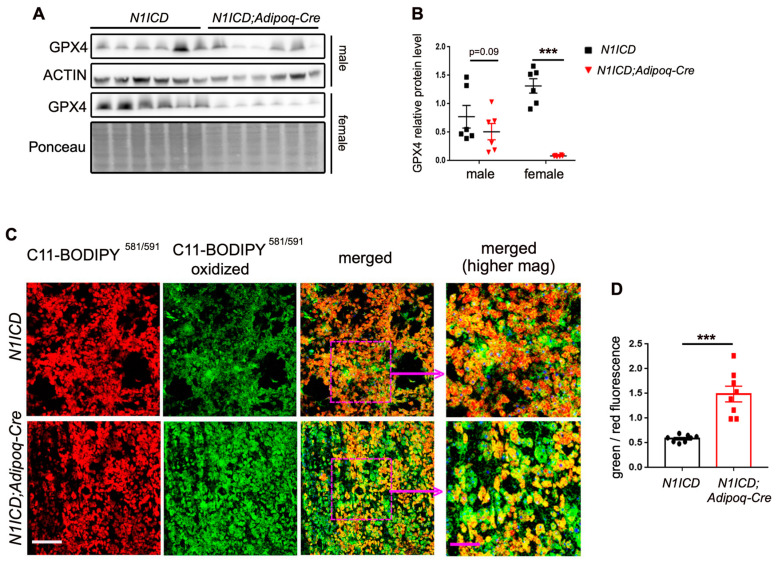
Activation of Notch signaling in PVAT-derived adipocytes correlates with increased ferroptosis. SVF isolated from PVAT of *N1ICD* or *N1ICD;Adipoq-Cre* mice were differentiated for 7 days to generate adipocytes (n = 6/group). (**A**) Immunoblotting was used to quantify GPX4 in differentiated cells from male or female mice. (**B**) Quantification of GPX4 normalized to actin or total protein determined using Ponceau staining. (**C**) Cells were stained with BODIPY™ 581/591 C11 to identify lipid peroxidation and ferroptosis. The ratio of fluorescent intensity between the green and red channels shows increased shifting of excitation and emission from 581/591 nm (red) to 488/510 nm (green) in cells from *N1ICD;Adipoq-Cre* mice. The boxed pink areas in the merged panels are presented at higher magnification in the right panels. White scale bar = 250 μm, and the pink scale bar in the higher magnification of the merged panels is 100 μm. (**D**) Fluorescence intensity was determined using ImageJ (version 1.53t) quantification of mean gray value of the green and red channel. n = 8 replicates from pooled cells were taken for each group. Graphed are means ± SEM. Statistical analysis was performed by Student’s *t*-test. ***: *p* < 0.001.

**Figure 6 genes-14-01964-f006:**
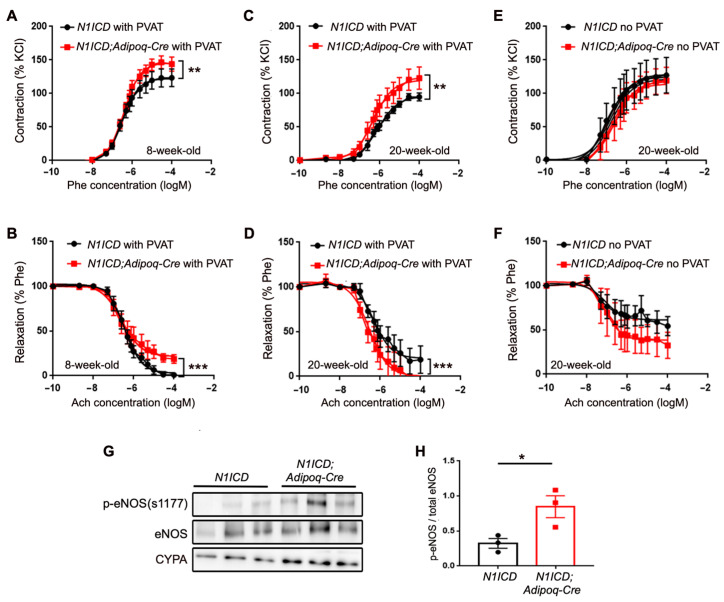
Notch signaling activation in PVAT leads to altered vascular function and eNOS-NO signaling. Thoracic aorta with PVAT from male *N1ICD* (n = 7) and *N1ICD;Adipoq*-*Cre* (n = 5) male mice at 8 weeks of age (**A**,**B**) and thoracic aorta with (**C**,**D**) or without PVAT (**E**,**F**) for *N1ICD* (n = 6) and *N1ICD;Adipoq*-*Cre* (n = 8) male mice at 20 weeks of age were examined for vasocontraction and vasorelaxation response using wire myography. Vasocontraction was tested using a dose response of phenylephrine (Phe) at 2 nM to 10 μM. Vasocontraction induced by Phe was normalized to contraction in response to 100 mM KCl. To test the vasodilation response, vessels were pre-contracted with Phe to ~50–70% maximal contraction and acetylcholine (Ach) administered at a dose range from 2 nM to 10 μM. Data are expressed as means ± SEM. Dose response curves were fitted using non-linear regression (four parameters). The logEC50 of each genotype was statistically compared using extra sum-of-squares F test, and statistical significance is displayed as *: *p* < 0.05; **: *p* < 0.01; ***: *p* < 0.001. (**G**) Total protein lysates were prepared from whole PVAT from 20 week old male *N1ICD* and *N1ICD;Adipoq-Cre* mice. Immunoblotting was performed to detect the phosphorylated form of eNOS and total eNOS. (**H**) p-eNOS/eNOS expression ratios were higher in the PVAT of *N1ICD;Adipoq-Cre* mice (red squares) compared to tissue from the *N1ICD* mice (black circles),. Data are graphed as means ± SEM. Statistical significance is displayed as *: *p* < 0.05, **: *p* < 0.01; ***: *p* < 0.001.

## Data Availability

The data are shared in PeptideAtlas (http://www.peptideatlas.org/PASS/PASS01599, accessed on 26 July 2020), and other data are available upon request.

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
