# Peer review of "Notch Signaling Regulates Mouse Perivascular Adipose Tissue Function via Mitochondrial Pathways"

_genes, 2023, doi:10.3390/genes14101964_

Round 1

Reviewer 1 Report

The research article looks interesting and talks about the importance of PVAT, its role in cardiovascular health, and the effects of obesity. The discussion about Notch signaling is clear and provides a good rationale for the study. My suggestions and comments are here.

·         The introduction is well-written and provides a clear rationale and background for the research. Minor adjustments in the framing of the sentence, as suggested below for example, could enhance the clarity and readability of the article.

Consider rewriting: "This is largely attributed to the obesity epidemic in the past decades." to "The rising prevalence of cardiovascular diseases (CVD) is largely attributed to the obesity epidemic observed over the past few decades."

·         The methods section has clear and comprehensive information on the tools used for data analysis, including software, statistical techniques, and visualization methods. Some improvements are required;

1)      While the breeding strategy for generating mice with Notch constitutive activation in adipose tissue is mentioned, more specifics on how the mice were selected after breeding, genotyping methods, or the genotypic ratios observed would have added clarity.

2)      Information on the total number of mice used, the number in each group, or how many experiments were replicated would provide a better context to understand the scope of the study.

·         Results:

1)      The description of the mouse models could be clearer. For instance, the difference between the "conditional N1ICD transgenic mice" and "N1ICD mice" is not immediately apparent to someone unfamiliar with this model. A brief one-line description of each might be helpful.

2)      The main topic seems to jump around from mitochondrial respiration to adipogenic differentiation, then back to mitochondrial biogenesis. A clearer organization will help guide the reader through the progression of experiments.

3)      Statements like "activation of Notch in the PVAT of male versus female mice yielded different PVAT phenotypes" require elaboration. What are these differences and why are they significant?

4)      How does the apparent sex-dependent effect of Notch signaling on mitochondrial metabolic function manifest in a physiological context? Are there potential implications for disease or metabolic disorders?

5)      Despite observed changes in mRNA levels for certain markers, no protein-level changes were observed. How is this discrepancy explained? What might be the regulatory or post-transcriptional processes at play?

6)      Repeated twice “homeostasis/metabolism phenotypes and mitochondrial phenotypes were altered by Notch activation in PVAT”, might be redundant for readers.

7)      How might alterations in mitochondrial dynamics (fusion and fission processes) due to Notch activation affect the overall cellular health of PVAT adipocytes?

8)      The connection between mitochondrial dysfunction, oxidative stress, and ferroptosis is stated without enough justification or references, which may leave readers unfamiliar with this connection slightly lost.

9)      The section starts with a focus on mitochondrial dysfunction leading to ferroptosis, but the connection between the two, especially in the context of Notch signaling, could be elaborated on more.

10)  Transitions between experiments could be smoother. For example, moving from the discussion about GPX4 protein levels to lipid peroxidation seems slightly abrupt.

11)  There's a potential inconsistency or typo in lines 561-562. The same figure, "Fig. 6E", is used to reference both the loss of differences with PVAT removed and the altered vascular function at 8 weeks. This could be confusing.

12)  The phrasing "Interesting, unlike the phenotype at 8 weeks of age" could be modified to "Interestingly, in contrast to the 8-week-old phenotype" for clarity.

13)  Would it be useful to investigate the vascular function in female mice or other age groups to get a more comprehensive understanding?

14)  Given the altered vascular function, are there potential therapeutic implications or strategies that could target Notch signaling in PVAT for vascular diseases?

·         Discussion:

1)      The discussion begins by presenting the broader context of PVAT's role, which is appropriate. However, the transition between discussing Notch signaling's role in metabolism and its upregulation in PVAT feels a bit abrupt.

2)      The ordering of some ideas could be improved. For example, mitochondrial dysfunction's impact on ferroptosis is discussed after the effects on mitochondrial pathways, which makes sense. However, the return to discussions on vascular function feels like backtracking.

3)      The role of Notch signaling in metabolic homeostasis is briefly mentioned, but there isn't much depth provided about the specifics of this relationship. This might benefit from a more in-depth analysis or referencing.

4)      The text mentions that the physiology of PVAT and its function significantly change during obesity but doesn't elucidate how Notch signaling might interact with obesity-driven changes.

5)      The authors provide numerous citations, indicating a well-researched discussion. However, it would be helpful to provide more context on how the present findings fit or diverge from the existing literature.

6)      The sentence "To further characterize the mechanism of Notch regulation of PVAT phenotype" feels incomplete.

7)      Some phrases like "These data all support the exciting premise..." are slightly editorialized. While the term "exciting" is subjective, it may be perceived as a lack of objectivity.

8)      It would be clearer to say something like, "These data collectively support the hypothesis...".

9)      There's a small grammatical inconsistency in "Thus physiology of PVAT...". It should probably be "The Physiology of PVAT..."

10)  There are a few instances of repetitive information, especially when talking about Notch overactivation's impact. For example, the mention of compromised anticontractile effects of PVAT seems to echo earlier statements in the discussion.

11)  The concluding section summarizes the study's main findings, which is good. However, it would be helpful to include future directions, potential clinical implications, or open questions the study has raised.

12)  How do you envision the potential therapeutic applications of these findings regarding Notch signaling and PVAT?

13)  Could the dysregulation of Notch signaling in PVAT be reversible, and if so, could reversing it offer therapeutic benefits?

14)  The discussion often refers to mouse models. How translatable are these findings to human physiology?

15)  Given that Notch signaling's role in metabolic homeostasis is mentioned, were there any observed systemic metabolic changes in mice with altered Notch signaling?

16)  Were there any observed differences in these effects between male and female mice, and if so, what might be driving these differences?

·         Supplementary Figure 2- c- What was the housekeeping for immunoblotting? Why supplementary figure 2 is in the middle of the manuscript? Figure S2 in original images-beta actin is very poor which doesn’t justify the PGC1alpha expression. All immunoblots' picture quality is poor.

·         Microscopy image quality is very poor.

·         Text quality in Figures 2 and 3 is very blurred and poor.

·         Conclusion is missing.

Too much repetition of the words and sentences. Grammar also needs to improve.

Reviewer 2 Report

The current manuscript aimed to investigate the notch signaling in regulation of mouse perivascular adipose tissue (PVAT) function via mitochondrial pathways. Using the Adiponectin-Cre (Adipoq-Cre) strain model, the authors showed that a ligand-independent Notch1 intracellular domain transgene (N1ICD) was activated to promote Notch signaling in adipose tissues (N1ICD;AdipoqCre). The authors used PVAT from Notch-activated versus control mice (N1ICD) for proteomic analysis, and they identified changes in cellular respiration and bio-energetics pathways. The authors also showed that adipocytes with Notch activation had significantly decreased mitochondrial respiration and mitochondrial biogenesis, and increased ferroptosis compared to controls, which was associated with increased mitophagy in PVAT with activation of Notch signaling. The current investigation suggested that that increased Notch signaling leads to PVAT whitening, impaired mitochondrial function, and loss of a protective vasodilatory signal. Overall, the experiments are properly performed and their data presented is clinically sound. However, some concerns listed below limit the clear narrative of the current study. 

1.     The Abstract should be revised for better organization and presentation to highlight their major findings and results in the manuscript.

2.     In the Statistical analysis session, the authors stated that: “Results were considered significantly different if the p-value was less than 0.5,”. I believe there's a typo in this statement that should be corrected to the number 0.05.

3.     The Supplemental Figures should be presented separated from the main Figures in the manuscript.

4.     In the Discussion Section, it is important for the reader to know what is limitation/weakness of the current study. The authors should comment, if possible, on potential limitations and weaknesses for this study, before concluding the discussion.

5.     There are some linguistic issues through the manuscript. The English language used in this manuscript should be improved before consideration of publication.

The English language used in this manuscript should be improved before consideration of publication.

Round 2

Reviewer 1 Report

The authors addressed all the comments.